# Scaling of an antibody validation procedure enables quantification of antibody performance in major research applications

Riham Ayoubi[1], Joel Ryan[2], Michael S Biddle[3], Walaa Alshafie[1], Maryam Fotouhi[1], Sara Gonzalez Bolivar[1], Vera Ruiz Moleon[1], Peter Eckmann[4], Donovan Worrall[1], Ian McDowell[1], Kathleen Southern[1], Wolfgang Reintsch[5], Thomas M Durcan[5], Claire Brown[2], Anita Bandrowski[4], Harvinder Virk[3], Aled M Edwards[6], Peter McPherson[1], Carl Laflamme[1]*

[1]Department of Neurology and Neurosurgery, Structural Genomics Consortium, The Montreal Neurological Institute, McGill University, Montreal, Canada; [2]Advanced BioImaging Facility (ABIF), McGill University, Montreal, Canada; [3]NIHR Respiratory BRC, Department of Respiratory Sciences, University of Leicester, Leicester, United Kingdom; [4]Department of Neuroscience, UC San Diego, La Jolla, United States; [5]The Neuro's Early Drug Discovery Unit (EDDU), Structural Genomics Consortium, McGill University, Montreal, Canada; [6]Structural Genomics Consortium, University of Toronto, Toronto, Canada

*For correspondence:
carl.laflamme@mcgill.ca

**Abstract** Antibodies are critical reagents to detect and characterize proteins. It is commonly understood that many commercial antibodies do not recognize their intended targets, but information on the scope of the problem remains largely anecdotal, and as such, feasibility of the goal of at least one potent and specific antibody targeting each protein in a proteome cannot be assessed. Focusing on antibodies for human proteins, we have scaled a standardized characterization approach using parental and knockout cell lines (Laflamme et al., 2019) to assess the performance of 614 commercial antibodies for 65 neuroscience-related proteins. Side-by-side comparisons of all antibodies against each target, obtained from multiple commercial partners, have demonstrated that: (*i*) more than 50% of all antibodies failed in one or more applications, (*ii*) yet, ~50–75% of the protein set was covered by at least one high-performing antibody, depending on application, suggesting that coverage of human proteins by commercial antibodies is significant; and (*iii*) recombinant antibodies performed better than monoclonal or polyclonal antibodies. The hundreds of underperforming antibodies identified in this study were found to have been used in a large number of published articles, which should raise alarm. Encouragingly, more than half of the underperforming commercial antibodies were reassessed by the manufacturers, and many had alterations to their recommended usage or were removed from the market. This first study helps demonstrate the scale of the antibody specificity problem but also suggests an efficient strategy toward achieving coverage of the human proteome; mine the existing commercial antibody repertoire, and use the data to focus new renewable antibody generation efforts.

## eLife assessment

Antibodies are some of the most important tools in biomedical research. However, their quality and specificity vary significantly. This **fundamental** study provides guidelines for how the quality of

an antibody should be assessed and recorded and provides **compelling** data on the selected anti-bodies. This paper will be of interest to researchers working in experimental cell biology.

## Introduction

Antibodies are critical reagents used in a range of applications, enabling the identification, quantification, and localization of proteins studied in biomedical and clinical research. The research enterprise spends significantly on the ~1.6 M commercially available antibodies targeting ~96% of human proteins (*Bandrowski et al., 2023*). Unfortunately, a significant percentage of these antibodies do not recognize the intended protein or recognize the protein but also recognize non-intended targets, with estimates that $0.375 to $1.75 billion is wasted yearly on non-specific antibodies (*Baker, 2015*; *Bradbury and Plückthun, 2015*; *Voskuil et al., 2020*). Perhaps worse, the use of poor-quality antibodies is a major factor in the scientific reproducibility crisis (*Bradbury and Plückthun, 2015*; *Voskuil et al., 2020*; *Baker, 2020*). With tens to hundreds of antibodies available for any given protein target, it is difficult for antibody users to select the best performing antibody (*Voskuil, 2014*), and a growing number of cases reveal that depending on previously published antibodies is not a reliable method to assess their performance (*Laflamme et al., 2019*; *Sato et al., 2021*; *Li et al., 2023*; *Sicherre et al., 2021*; *Haytural et al., 2019*; *Virk et al., 2019*). Academic and industry scientists aspire to have at least one, and ideally more, potent, selective and renewable antibody for each human protein for each of the most common applications (*Marx, 2020*). Unfortunately, there is no agreed-upon mechanism to determine, validate or compare antibody performance and there are multiple strategies for antibody validation (*Uhlen et al., 2016*), with unequal scientific value. Most information on how commercial antibodies perform is anecdotal. It is thus difficult to assess progress toward the objective of well-validated antibodies for each human protein, or to design a strategy to accomplish this aim.

We sought to address this issue by developing optimized protocols to assess antibody specificity in the three most common uses of antibodies in biomedical research laboratories; Western blot (WB), immunoprecipitation (IP), and immunofluorescence (IF). We used these protocols to test antibodies against a variety of neuroscience targets, chosen by funders, to predict requirements for the larger goal of coverage of an entire mammalian proteome. The optimal antibody testing methodology is largely settled; using an appropriately selected wild type cell and an isogenic CRISPR knockout (KO) version of the same cell as the basis for testing, yields rigorous and broadly applicable results (this study, as well as *Laflamme et al., 2019*; *Ellis et al., 2023*; *Davies et al., 2013*). However, the cost of antibody characterization using engineered KO cells is higher than that of other methods, mainly because of the cost of custom edited cells. Commercial antibody suppliers support a large and diverse catalogue of products, with most antibody products generating <$5000 in total sales, far less than the costs of KO-based validation, estimated at $25,000. While leading companies are increasingly assessing antibody performance, it is exceedingly difficult, and cost restrained, to properly characterize all their products. Even when available, high-performing antibodies may remain hidden within the millions of reagents of unknown quality.

To begin the process of large-scale antibody validation and to provide a large enough dataset to allow for more accurate estimates of the work and financing required to complete such a project, we began with the human proteome. We created a partnership of academics, funders, and commercial antibody manufacturers, including 10 companies representing approximately 27% of antibody manufacturing worldwide. For each protein target, we tested commercial antibodies, provided from various manufacturers, in parallel using standardized protocols, agreed upon by all parties, in WB, IP, and IF applications. All data are shared rapidly and openly on ZENODO, a preprint server. We have tested 614 commercially available antibodies targeting 65 proteins, and found that approximately two thirds of this protein set was covered by at least one high-performing antibody, and half was covered by at least one high-performing renewable antibody, suggesting that coverage of human proteins by high-performing antibodies is significant. This sample is large enough to observe several trends in antibody performance across various parameters and estimate the scale of the antibody liability crisis.

**eLife digest** Commercially produced antibodies are essential research tools. Investigators at universities and pharmaceutical companies use them to study human proteins, which carry out all the functions of the cells. Scientists usually buy antibodies from commercial manufacturers who produce more than 6 million antibody products altogether. Yet many commercial antibodies do not work as advertised. They do not recognize their intended protein target or may flag untargeted proteins. Both can skew research results and make it challenging to reproduce scientific studies, which is vital to scientific integrity. Using ineffective commercial antibodies likely wastes $1 billion in research funding each year.

Large-scale validation of commercial antibodies by an independent third party could reduce the waste and misinformation associated with using ineffective commercial antibodies. Previous research testing an antibody validation pipeline showed that a commercial antibody widely used in studies to detect a protein involved in amyotrophic lateral sclerosis did not work. Meanwhile, the best-performing commercial antibodies were not used in research. Testing commercial antibodies and making the resulting data available would help scientists identify the best study tools and improve research reliability.

Ayoubi et al. collaborated with antibody manufacturers and organizations that produce genetic knock-out cell lines to develop a system validating the effectiveness of commercial antibodies. In the experiments, Ayoubi et al. tested 614 commercial antibodies intended to detect 65 proteins involved in neurologic diseases. An effective antibody was available for about two thirds of the 65 proteins. Yet, hundreds of the antibodies, including many used widely in studies, were ineffective. Manufacturers removed some underperforming antibodies from the market or altered their recommended uses based on these data. Ayoubi et al. shared the resulting data on Zenodo, a publicly available preprint database. The experiments suggest that 20-30% of protein studies use ineffective antibodies, indicating a substantial need for independent assessment of commercial antibodies.

Ayoubi et al. demonstrated their side-by-side antibody comparison methods were an effective and efficient way of validating commercial antibodies. Using this approach to test commercial antibodies against all human proteins would cost about $50 million. But it could save much of the $1 billion wasted each year on research involving ineffective antibodies. Independent validation of commercial antibodies could also reduce wasted efforts by scientists using ineffective antibodies and improve the reliability of research results. It would also enable faster, more reliable research that may help scientists understand diseases and develop new therapies to improve patient's lives.

## Results

### Assembling KO cell lines and antibodies

Our initiative has thus far validated antibodies for 65 protein targets, which were chosen by disease charities, academia, and industry without consideration of antibody coverage. The list is comprises 32 Alzheimer's disease (AD)-related proteins that were community-nominated through an NIH-funded project on dark AD genes (https://agora.adknowledgeportal.org/), 22 proteins nominated within the amyotrophic lateral sclerosis (ALS) Reproducible Antibody Platform project, 5 Parkinson's disease (PD)-linked proteins nominated by the Michael J. Fox Foundation, and 6 proteins nominated by industry (*Figure 1A*). Within the 65 target proteins, 56 are predicted intracellular and 9 are predicted secreted. The description of each protein target is indicated in *Figure 1—source data 1*.

The proteins were searched to determine the Uniprot identifier, the predicted molecular mass, and whether the protein is secreted or intracellular (*Figure 1B*). Our strategy was predicated on identifying a parental cell line that expressed sufficient levels of the target protein to be detected by an antibody with a binding affinity of 1–50 nM. To identify candidate lines, we searched the Cancer Dependency Map Portal (DepMap) using the 'Expression 22Q1' database, which houses the RNA-level analysis of >1800 cancer cell lines (*Ghandi et al., 2019*; *Figure 1B*). After our initial experience with a few dozen targets comparing RNA expression and the ability to detect a clear signal, we selected 2.5 $\log_2$(TPM +1) as an RNA-level threshold to select a candidate cell line to create a KO. Among the cell lines showing expression above this level, we prioritized a group of 8 common cell line backgrounds

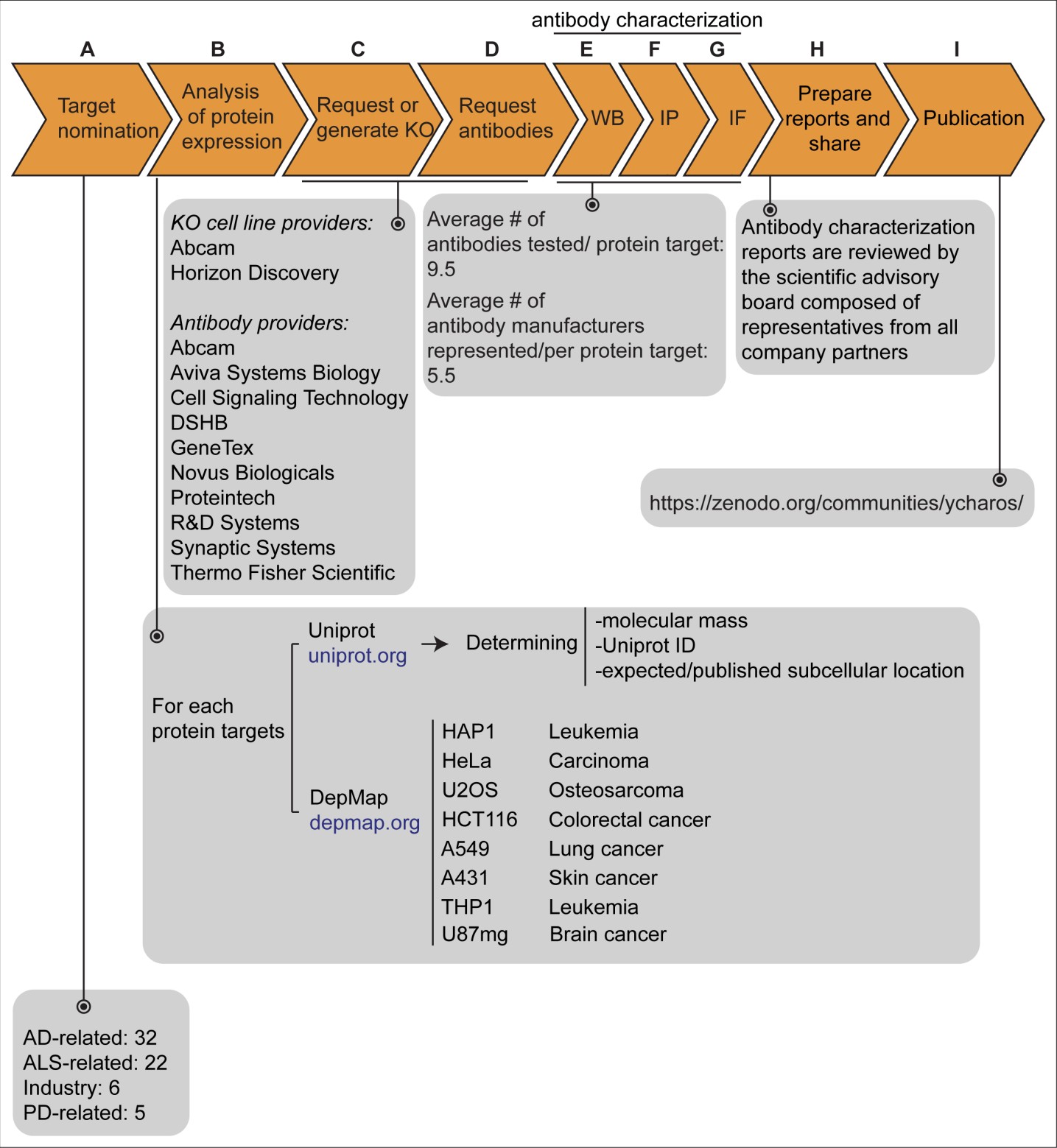

**Figure 1.** Antibody characterization platform. (**A**) The funders of the targets analyzed in this study and the number of targets proposed by each are indicated. (**B**) Bioinformatic analyses of nominated proteins using Uniprot to determine their molecular mass, unique Uniprot ID and published/expected subcellular distribution. In parallel, analyses of the Cancer Dependency Map ('DepMap') portal provided RNA sequencing data for the designated target, which guided our selection of cell lines with adequate expression for the generation of custom KO cell lines. A subset of cell lines amenable for genome engineering were prioritized. (**C**) Receive relevant KO cell lines or generate custom KO lines and (**D**) receive antibodies from manufacturing partners. All contributed antibodies were tested in parallel by (**E**) WB using WT and KO cell lysates ran side-by-side, (**F**) IP followed by WB using a KO-

*Figure 1 continued on next page*

Figure 1 continued

validated antibody identified in (**E**) and by (**G**) IF using a mosaic strategy to avoid imaging and analysis biases. (**H**) Antibody characterization data for all tested antibodies were presented in a form of a protein target report. All reports were shared with participating companies for their review. (**I**) Reviewed reports were published on ZENODO, an open access repository. ALS-RAP=amyotrophic lateral sclerosis-reproducible antibody platform, AD = Alzheimer's disease, MJFF = Michael J. Fox Foundation. KO = knockout cell line.

The online version of this article includes the following source data and figure supplement(s) for figure 1:

**Source data 1.** Description of the 65 nominated target proteins.

**Figure supplement 1.** Schematic representations of antibody performance.

representing different cell/tissue types because their doubling time is short, and they are amenable to CRISPR-Cas9 technology (*Figure 1B*). These 8 cell lines were used in 62 out of the 65 antibody characterization studies (*Figure 1—source data 1*).

After identifying candidate cell lines for each target, we either obtained KO lines from our industry consortium partners or generated them in-house (*Figure 1C*). Antibodies were provided from antibody manufacturers, who were responsible for selecting antibodies to be tested from their respective catalogs (*Figure 1D*). Most antibody manufacturers prioritized renewable antibodies. The highest priority was given to recombinant antibodies as they represent the ultimate renewable reagent (*Marx, 2020*) and have advantages in terms of adaptability, such as switching IgG subclass (*Andrews et al., 2019*) or using molecular engineering to achieve higher affinity binding than B-cell generated antibodies (*Gray et al., 2020*).

All available antibodies from all companies were tested side-by-side in parental and KO lines. The protocols used were established by our previous work (*Laflamme et al., 2019*) and refined in collaboration with antibody manufacturers. On occasion, our protocols differed from those the companies used in their internal characterization. All antibodies were tested for all three applications (except that secreted proteins were not tested in IF), independent of the antibody manufacturers' recommendations. We received on average 9.5 antibodies per protein target contributed from an average of five different antibody manufacturers (*Figure 1E, F and G*). Companies often contributed more than one antibody per target (*Figure 1—source data 1*).

## Antibody and cell line characterization

For WB, antibodies were tested on cell lysates for intracellular proteins or cell media for secreted proteins (*Figure 1E*). For 55/65 of the target proteins, we identified one or more antibodies that successfully immunodetected their cognate protein, identifying well-performing antibodies and validating the efficacy of the KO lines. For the remaining nine targets, we identified at least one specific, non-selective antibody that detects the cognate protein by WB, but also recognizes unrelated proteins, that is, non-specific bands not lost in the KO controls. All 614 antibodies were tested by IP on non-denaturing cell lysates for intracellular proteins or cell media for secreted proteins, using WB with a successful antibody from the previous step to evaluate the immunocapture (*Figure 1F*). All antibodies against intracellular proteins were tested for IF using a strategy that imaged a mosaic of parental and KO cells in the same visual field to reduce imaging and analysis biases (*Figure 1G*).

For each protein target, we consolidated all screening data into a report, which is made available without restriction on ZENODO, a data-sharing website operated by CERN. On ZENODO, all 65 reports are gathered under the Antibody Characterization through Open Science (YCharOS) community: https://ZENODO.org/communities/ycharos/ (*Figure 1I*). Prior to release, each antibody characterization report underwent technical peer review by a group of scientific advisors from academia and industry (*Figure 1H*).

## Coverage of human proteins by renewable antibodies

The Antibody Registry (https://www.antibodyregistry.org, RRID:SCR_006397) indicates that there are ~1.6 million antibodies covering ~96% of human proteins (*Bandrowski et al., 2023*), with 53% covered by at least five renewable antibodies (*Figure 2A*, *Figure 2—source data 1*). Approximately 21% of human proteins are covered by only one or two renewable antibodies, and ~15% have no renewable antibodies available (*Figure 2A*). In our set of 65 proteins, and from the manufacturers

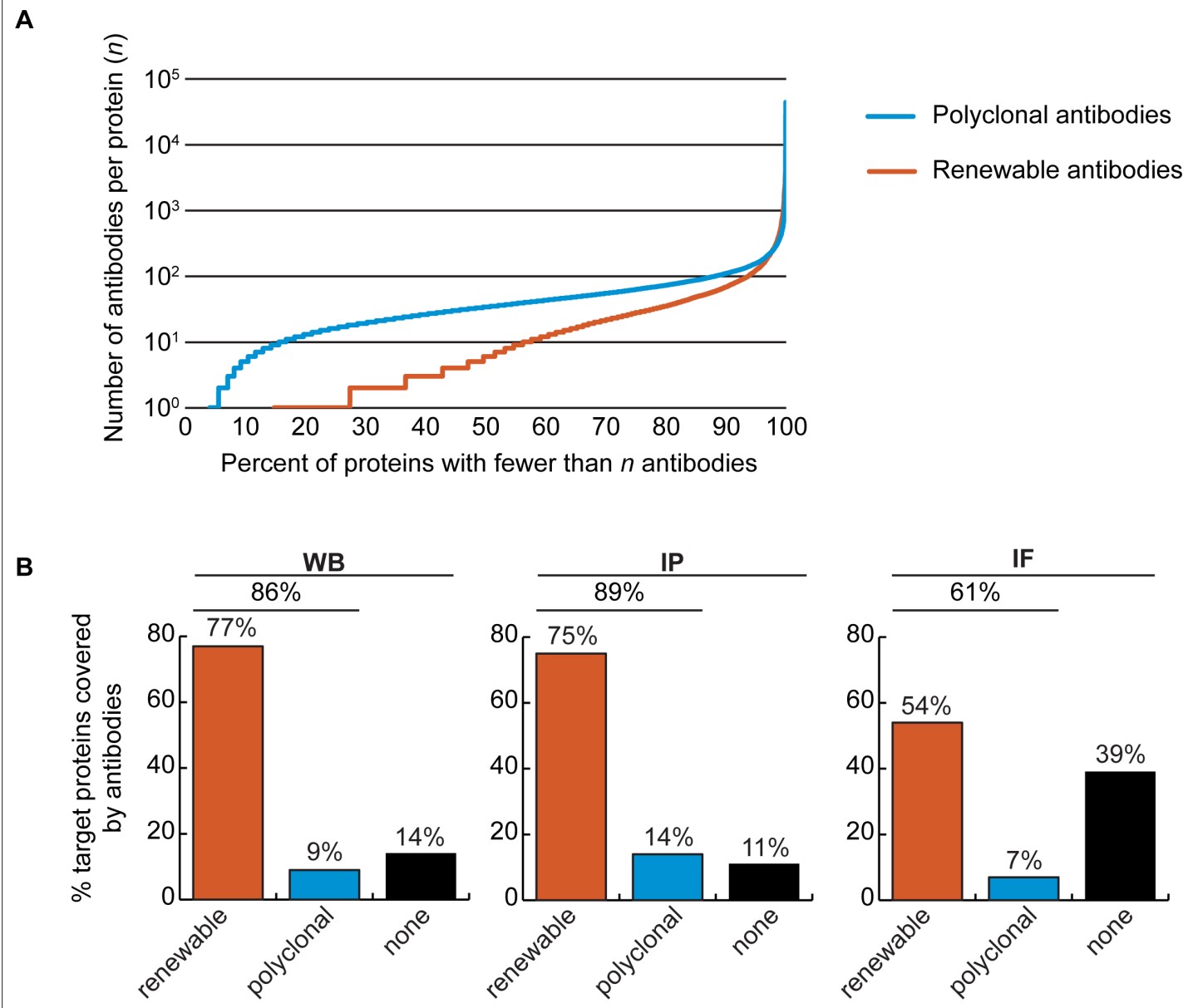

**Figure 2.** Analysis of human protein coverage by antibodies. (**A**) Cumulative plot showing the percentage of the human proteome that is covered by polyclonal antibodies (blue line) and renewable antibodies (monoclonal +recombinant; orange line). The number of antibodies per protein was extracted from the Antibody Registry database. (**B**) Percentage of target proteins covered by minimally one renewable successful antibody (orange column) or covered by only successful polyclonal antibodies (blue column) is shown for each indicated applications using a bar graph. Lack of successful antibody ('none') is also shown (black column).

The online version of this article includes the following source data for figure 2:

**Source data 1.** Number of antibodies per human protein.

represented, 49 were covered by at least 3 renewable antibodies, 15 by 1 or 2 renewable antibodies, and 1 was not covered by any renewables (*Figure 1—source data 1*).

We found a well-performing renewable antibody for 50 targets in WB (*Figure 2B*, left bar graph), for 49 targets in IP (*Figure 2B*, middle bar graph), and for 30 targets in IF (*Figure 2B*, right bar graph). For some proteins lacking coverage by renewable antibodies or lacking successful renewable antibodies, well-performing polyclonal antibodies were identified (*Figure 2B*). Some proteins were not covered by any successful antibodies depending on application; notably ~40% of our protein set lacked a successful antibody for IF (*Figure 2B*, right bar graph).

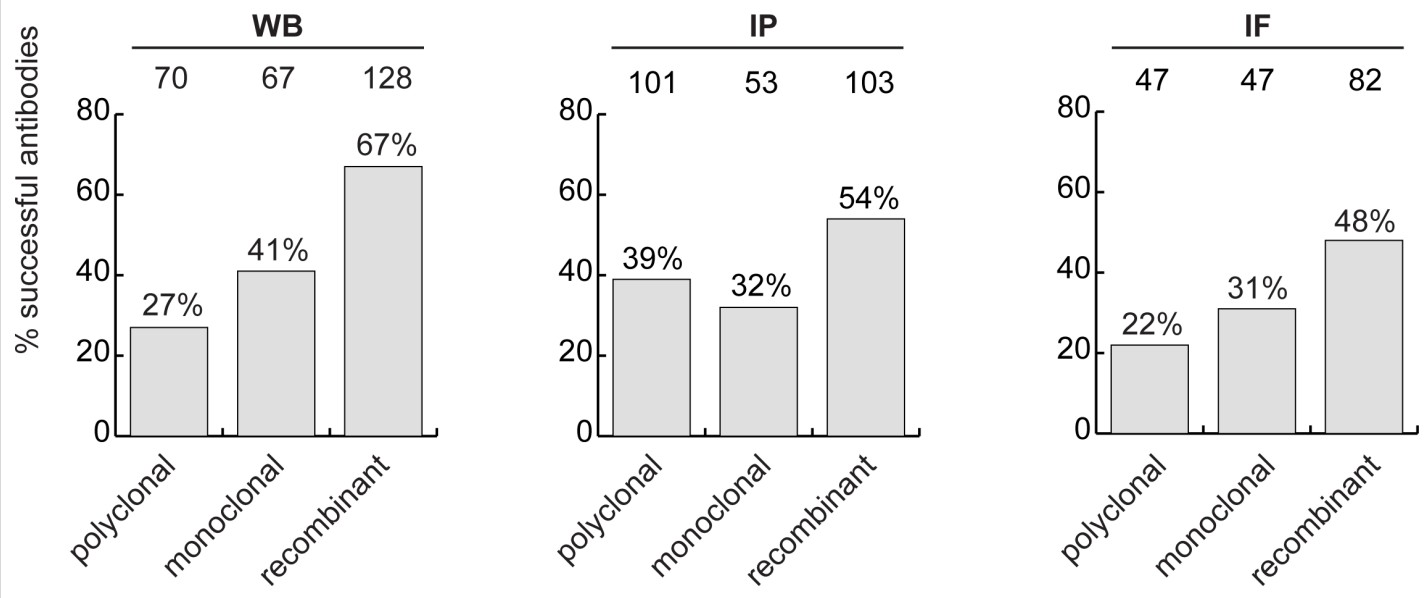

**Figure 3.** Analysis of antibody performance by antibody types. The percentage of successful antibodies based on their clonality is shown using a bar graph, for each indicated application. The number of antibodies represented in each category is indicated above the corresponding bar.

The online version of this article includes the following figure supplement(s) for figure 3:

**Figure supplement 1.** Correlation of antibody performance between applications.

## Recombinant antibody performance

The antibody set constituted 258 polyclonal antibodies, 165 monoclonal antibodies and 191 recombinants. For WB, 27% of the polyclonal antibodies, 41% of the monoclonal antibodies and 67% of the recombinant antibodies immunodetected their target protein (*Figure 3*, left bar graph). For IP, trends were similar: 39%, 32% and 54% of polyclonal, monoclonal and recombinants, respectively (*Figure 3*, middle bar graph). For IF, we tested 529 antibodies against the set of intracellular proteins; 22% of polyclonal antibodies, 31% of monoclonal antibodies, and 48% of recombinant antibodies generated selective fluorescence signals in images of parental versus KO cells (*Figure 3*, right bar graph). Thus, recombinant antibodies are on average better performers than polyclonal or monoclonal antibodies in each of the applications. It should be noted that recombinant antibodies are newer protein reagents compared to polyclonal and monoclonal hybridomas, and their superior performance could be a consequence of enhanced internal characterization by the commercial suppliers.

Our analyses also inform the characterization pipelines to use for newly generated renewable antibodies. Currently, it is common to use WB as the initial screen (*Lund-Johansen and Browning, 2017*). However, we find that success in IF is the best predictor of performance in WB and IP (*Figure 3— figure supplement 1*).

## Optimizing an antibody characterization strategy

While the parental versus KO method is the consensus superior method for antibody validation (*Laflamme et al., 2019*; *Ellis et al., 2023*; *Davies et al., 2013*; *Lutz et al., 2022*), not all antibodies on the market are characterized this way, largely due to cost and the range of alternative methods (*Uhlen et al., 2016*). To assess if the cost of KO characterization is justified, we compared the performance of antibodies in our dataset to the performance predicted by the characterization methods used by the companies. In all, 578 of the 614 antibodies tested were recommended for WB by the manufacturers. Of these, 44% were successful, 35% were specific but non-selective, and 21% failed (*Figure 4—figure supplement 1*, left bar graph). Most antibodies are not recommended for IP by the suppliers, perhaps because they are not tested. Of 614 antibodies, 143 were recommended for IP, and 58% enriched their cognate target from cell extracts. Interestingly, of the 471 remaining antibodies that had no recommendation for IP, 37% were able to enrich their cognate antigen (*Figure 4—figure supplement 1*, middle bar graph). In this regard, the manufacturers are not sufficiently recommending their

successful products. Of the 529 antibodies tested in IF, 293 were recommended for this application by the suppliers and 236 were not. Only 39% of the antibodies recommended for IF were successful (*Figure 4—figure supplement 1*, right bar graph).

We next investigated if antibody validation strategies have equal scientific value. Broadly, antibodies are characterized using genetic approaches, which exploit KO or knockdown (KD) samples as controls, or using orthogonal approaches, which rely on known information about the target protein of interest as a correlate to validate performance. For WB, 61% of antibodies were recommended by manufacturers based on orthogonal approaches, 30% based on genetic approaches and 9% using other strategies. For IF, 83% of the antibodies were recommended based on orthogonal approaches, 7% using genetic approaches and 10% using other strategies (*Figure 4A*). For WB, 80% of the antibodies recommended by the manufacturers based on orthogonal strategies and 89% of antibodies recommended based on genetic strategies could detect the intended target protein (*Figure 4B*, left bar graph). For IF, 38% of the antibodies recommended by the manufacturers based on orthogonal strategies were confirmed using KO cells as controls. Of the 20 antibodies validated by the manufacturers for IF on the basis of genetic strategies, we confirmed the performance of 16 (80%) (*Figure 4B*, histogram right). Of the four antibodies that failed in our hand, one has already been withdrawn from the market by the manufacturer. Thus, while orthogonal strategies are somewhat suitable for WB, genetic strategies generate far more robust characterization data for IF.

From a total of 409 antibodies that presented conflicting data between our characterization data and antibody supplier's recommendations, the participating companies have withdrawn 73 antibodies from the market and changed recommendations for 153 antibodies (*Figure 4—figure supplement 2*). In turn, high-quality antibodies are being promoted. We expect to see additional changes and an overall improvement in the general quality of commercial reagents as more antibody characterization reports are generated.

## Antibodies and reproducible science

The availability of renewable, well-characterized antibodies would be expected to enhance the reproducibility of research. To assess the bibliometric impact of underperforming antibodies, we used the reagent search engine CiteAb (https://www.citeab.com/) to quantify how antibodies in our dataset have been used in the literature. We identified 2010 publications that employed one of the 180 antibodies we tested for WB. Of those, 69% used a well-performing antibody that specifically immunodetected its target protein by WB, while 31% used an antibody unsuccessful in our protocol (*Figure 4C*). For IP, 105 publications employed 41 of our tested antibodies while 65% of these used a well-performing antibody but 35% employed an antibody unable to immunocapture its target protein (*Figure 4C*). For IF, we found 548 publications that employed 80 of the antibodies we tested. Of these publications, 22% used an antibody unable to immunolocalize its target protein (*Figure 4C*), with 88% containing no validation data (*Figure 4D*). If our results are representative, this suggests that 20–30% of figures in the literature are generated using antibodies that do not recognize their intended target, and that more effort in antibody characterization is highly justified.

A Research Resource Identification (RRID) was assigned to each of the 614 antibodies tested, indicated in each of the 65 antibody characterization reports available on ZENODO (*Figure 5*, bottom right image). Antibody characterization data generated by this organization are being disseminated by the RRID community and are directly connected through the Antibody Registry, or the RRID Portal (*Figure 5*, bottom left image) and participating antibody manufacturers' websites (*Figure 5*, top image).

## Discussion

Here, we present the analysis of a dataset of commercial antibodies as an assessment of the problem of antibody performance, and as a step toward a comprehensive and standardized ecosystem to validate commercial antibodies. We evaluated 614 antibodies against 65 human proteins side-by-side in WB, IP, and IF. All raw data are openly available (https://ZENODO.org/communities/ycharos/), identifiable on the RRID portal and on participating antibody manufacturers' websites. Our studies provide an unbiased and scalable analytical framework for the representation and comparison of antibody performance, an estimate of the coverage of human proteins with renewable antibodies, an

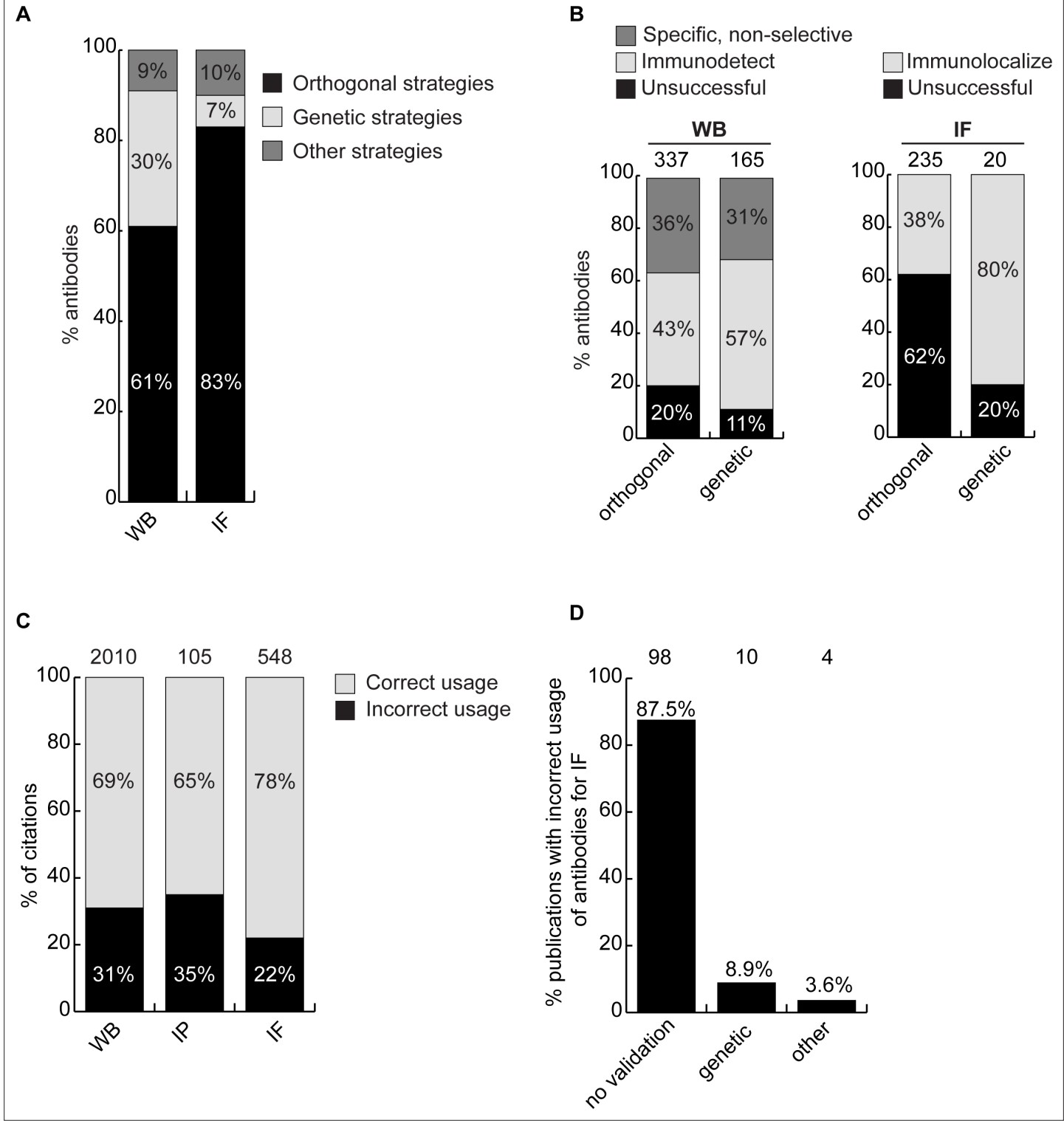

**Figure 4.** Scientific value of antibody characterization methods and research usage. (**A**) Percentage of antibodies validated by suppliers using one of the indicated methods for WB or IF showed using a bar graph with stacked columns. The percentage corresponding to each section of the bar graph is shown directly in the bar graph. Orthogonal = orthogonal strategies, genetic = genetic strategies. (**B**) Percentage of successful (light gray), specific, non-selective (dark gray-only for WB) and unsuccessful (black) antibodies according to the validation method used by the manufacturer for WB and IF as compared to the KO strategy used in this study. Data are shown using a bar graph with stacked columns. The percentage corresponding to each section of the bar graph is shown directly in the bar graph. The number of antibodies analyzed corresponding to each condition is shown above each bar. (**C**) Percentage of publications that used antibodies that successfully passed validation (correct usage) or to antibodies that were unsuccessful

*Figure 4 continued on next page*

*Figure 4 continued*

in validation (incorrect usage) showed using a bar graph with stacked columns. The number of publications was found by searching CiteAb. The percentage corresponding to each section of the bar graph is shown in the bar graph and the number of publications represented in each category is shown above the corresponding bar. (**D**) Percentage of publications that used an unsuccessful antibody for IF from (**C**) that provided validation data for the corresponding antibodies. Data is shown as a bar graph. The number of publications represented in each category is shown above the corresponding bar.

The online version of this article includes the following figure supplement(s) for figure 4:

**Figure supplement 1.** Analysis of antibody performance by manufacturer's catalogue recommendation.

**Figure supplement 2.** Actions taken from participating companies.

assessment of the scientific value of common antibody characterization methods, and they inform a strategy to identify renewable antibodies for all human proteins.

Our approach, developed in collaboration with manufacturers, and intended to be applied to entire proteomes, uses universal protocols for all tested antibodies in each application. Scientists use variants of such protocols, optimized for their protein of interest, which can have a major impact on antibody performance (*Pillai-Kastoori et al., 2020*; *Piña et al., 2022*; *Marcon et al., 2015*). Nevertheless, the process robustly identifies antibodies that fail to recognize their intended target, which becomes evident when other antibodies tested in parallel perform well. At a minimum, removal of these poorly performing products from the market will have significant impact in that hundreds of published papers report the use of such antibodies.

The impacts of poorly performing antibodies are well documented (*Voskuil et al., 2020*; *Sato et al., 2021*; *Aponte Santiago et al., 2023*; *Freedman et al., 2016*); our analyses provide insight

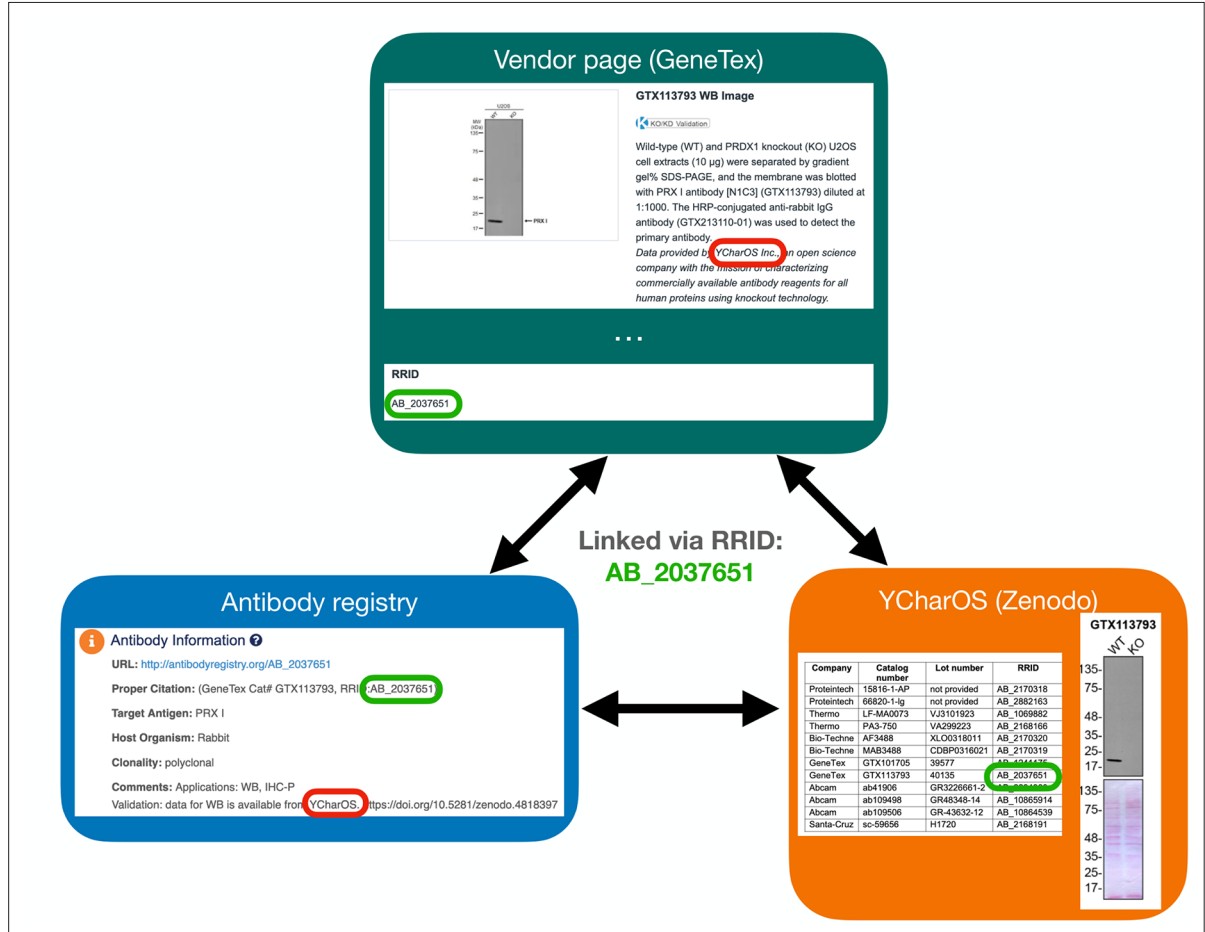

**Figure 5.** Accessing antibody characterization data using RRIDs. An antibody RRID can be used to search characterization studies across various databases, such as the vendor page, the Antibody Registry and on the YCharOS community page on ZENODO. AB_2037651 is given as an example.

into the magnitude of the problem. In our set of 65 proteins, we found that an average of ~12 papers per protein included use of an antibody that failed to recognize the intended protein target using our protocols. Scientists are not entirely to blame; dozens of antibodies can be used in a single study, often unrelated to the authors' protein of interest. Genetic validation of every antibody used in a study remains a difficult, if not impossible task. In addition, even with our optimized protocol, the cost of characterizing antibodies for a single protein is estimated at ~$25,000 USD. And if each investigator performs such an analysis, there will be multiple overlapping validation of any given antibody. We estimate a cost of $50 million USD to characterize antibodies against all proteins in a proteome, considering parallelization and industrialization of the procedure. The costs mentioned exclude the expenses of antibodies and knockout cell lines. However, it should be noted that this estimated cost for validation is far below the predicted waste on bad antibodies, currently estimated at ~$1B/year (*Voskuil et al., 2020*). Thus, independent antibody characterization with openly published data, funded by various global organizations, is an important, if not essential, initiative that is certain to save large amounts of money and increase the quality and reproducibility of the literature. This study demonstrates the feasibility of such an initiative.

Life scientists tend to focus on a small subset of human proteins, leading to an imbalance between a small percentage of well-studied proteins, and a higher percentage of poorly characterized proteins (*Carter et al., 2019*). Our set of 65 funder-designated proteins is an unbiased sample, representative of the heterogeneity of knowledge of the human proteome; a search of the NIH protein database revealed that 15 proteins (23% of our protein sample) are well studied with more than 500 publications and 50 proteins (77% of our protein sample) have corresponding publications ranging from 37 to 498 (*Figure 1—source data 1*). Although we observed that there are more commercial antibodies available for the best-studied proteins (*Figure 1—source data 1*), an encouraging result of our work is that more than half of our protein targets are covered by well-performing, renewable antibodies for WB, IP and IF - including both well characterized and more poorly studied proteins. Within the antibodies we tested, we found a successful renewable antibody for WB for 77% of proteins (50/65), for IP for 75% of proteins (49/65) and for IF for 54% of proteins (30/56) examined. Extrapolation of our findings to the human proteome would suggest that it might be possible to identify well-performing renewable reagents for half the human proteome, including poorly characterized proteins, simply by mining commercial collections. Indeed, it is likely that the coverage is greater because our corporate partners only represent 27% of the antibody production worldwide.

The research market is heavily dominated by polyclonal antibodies, and their use contributes to reproducibility issues in biomedical research (*Baker, 2015*; *Baker, 2020*) and present important ethical concerns. From a scientific perspective, polyclonal antibodies suffer from batch-to-batch variation and are thus in conflict with the scientific community desire to use and provide only renewable reagents. From an ethical perspective, the generation of polyclonal antibodies requires large numbers of animals yearly (*Gray et al., 2016*). While recombinant antibodies may rely on the use of animals for the initiation of an antibody generation program, animal-free in vitro molecular strategies are also used for production, and to generate new batches of these antibodies (*Gray et al., 2020*). As of today, the uptake of recombinant antibodies by the scientific community has not been satisfactory. For example, while leading antibody manufacturers are converting top-cited polyclonal antibodies into recombinant antibodies and removing underperforming antibodies from their catalogues, polyclonals remain the most purchased. This situation has also been acknowledged by the EU Reference Laboratory for Alternatives to Animal Testing and a lack of understanding in the use of recombinant methods has been suggested by authors of a recent correspondence to the editors of Nature Biotechnology (*Gray et al., 2020*). One reason for this confusion could be the absence of large-scale performance data comparing the various antibody generation technologies. In our dataset, recombinant antibodies performed well in all applications tested, arguing there is no reason not to adopt the recombinant technology. Moreover, our study strongly supports the idea that future antibody generation programs should focus on recombinant technologies.

Our analysis of antibody performance indicates that success in IF is an excellent predictor of performance in WB and IP. Given that it is difficult to imagine a characterization pipeline dependent on IF, we suggest that using KO (or knockdown in the case of an essential gene) strategies to screen antibodies for the intended application will provide the most effective approach to identify selective antibodies. Currently, one of the main barriers to large-scale production of high-quality antibodies

is the lack of availability of KO lines derived from cells that express detectable levels of each human protein. Creation of a broadly accessible biobank of bespoke KO cells for each human gene should be a priority for the community.

Our studies are rapidly shared via the open platform ZENODO, and selected studies were published on the F1000 publication platform (https://f1000research.com/ycharos). This data generation and dissemination is intended to benefit the global life science community, but its impact depends on the real-world uptake of the data. In addition, we recognize that antibodies are used in other protocols or in variations of our protocols that may yield important new or different outcomes. Posting of such information from users worldwide on open platforms will allow continued improvements to the data. Thus, we have partnered with the RRID Portal Community to improve our dissemination strategies. The Antibody Registry is a comprehensive repository of over 2.5 million commercial antibodies that have been assigned with RRIDs to ensure proper reagent identification (*Bandrowski et al., 2023*). Our data can be searched in the AntibodyRegistry.org and other portals that display this data such as the RRID.site portal and dkNet.org. The search term 'ycharos' will return all the currently available antibodies that have been characterized and searching for the target or the catalogue number of the antibody in any of these portals will also bring back the YCharOs information. In the RRID.site portal and dkNet there will also be a green star, tagging the antibodies to further highlight the contribution of YCharOS. The project is also being promoted through large international bioimaging networks including Canada BioImaging (CBI - https://www.canadabioimaging.org/), BioImaging North America (BINA - https://www.bioimagingnorthamerica.org/) and Global BioImaging (GBI - https://globalbio-imaging.org/).

Overall, this project provides the global life sciences community with a tremendous resource for the study of human proteins and will result in significant improvements in rigour and reproducibility in antibody-based assays and scientific discovery.

# Materials and methods

## Data analysis

Performance of each antibody was retrieved from the corresponding ZENODO report or publication (*Figure 1—source data 1*), for WB, IP and IF, and analyzed following the performance criteria described in *Table 1*. Antibody properties, application recommendations and antibody characterization strategies were taken from the manufacturers' datasheets. Throughout the manuscript, renewable antibodies refer to monoclonal antibodies from hybridomas and to recombinant antibodies (monoclonal and polyclonal recombinant antibodies) generated in vitro.

For *Figure 2A*, the analysis of the antibody coverage of human proteins was performed as previously described (*Bandrowski et al., 2023*) and antibodies were divided into polyclonal and renewable categories.

To evaluate the number of citations corresponding to each tested antibody (*Figure 4C*), we searched CiteAb (between November 2022 and March 2023) and used the provided analysis of citations per application. We then searched for publications mentioning the use of a poorly performing antibody for IF. Publications were filtered by application (ICC, ICC-IF, and IF) and reactivity (*Homo*

**Table 1.** Antibody performance criteria.

| | Definition |
|---|---|
| Successful antibody for western blot | A successful primary antibody immunodetects the target protein, and the signal observed in the WT lysate is lost in the KO lysates (*Figure 1—figure supplement 1A*). The antibody does not recognize other proteins under the conditions tested. |
| Specific, non-selective antibody for western blot | The primary antibody specifically recognizes the target protein, but also unrelated protein(s) (*Figure 1—figure supplement 1A*). |
| Successful antibody for immunoprecipitation | Under the conditions used, a successful primary antibody immunocaptures the target protein to at least 10% of the starting material (*Figure 1—figure supplement 1B*). |
| Successful antibody for immunofluorescence | A successful primary antibody immunolocalizes the target protein by generating a fluorescence signal in WT cells that is at least 1.5-fold higher than the signal in KO cells (*Figure 1—figure supplement 1C*). Signal provided by such antibody staining can be easily distinguished from unspecific background and noise. |

*sapiens*) on CiteAb (on July 2023), each publication being manually checked to confirm antibody and technique. This resulted in 112 publications, which were then assessed for characterization data (*Figure 4D*).

We asked participating antibody suppliers to indicate the number of antibodies eliminated from the market, and the number of antibodies for which there was a change in recommendation due to their evaluation of our characterization data (*Figure 4—figure supplement 2*).

The correlation of antibody performance between two applications were evaluated by the McNemar test, followed by the chi-square statistic (*Figure 3—figure supplement 1*). The number of antibodies was reported in each corresponding cell of the 2x2 contingency tables, and chi-square statistic was computed as follows: $X2 = (b - c) \, 2/b + c$. The null hypothesis is $p_b = p_c$ (where p is the population proportion). Note that these hypotheses relate only for the cells that assess change in status, that is cell b which contains the number of antibodies which passed application #2, but failed application #1, whereas cell c contains the number of antibodies which passed application #1, but failed application #2. The test measures the effectiveness of antibodies for one application (from fail to pass) against the other application (change from pass to fail). If $p_b = p_c$, the performance of one application is not correlated with the performance of another application, whereas if $p_b <$ or $>_{pC}$, then antibody performance from one application can inform on the performance of the other application. The computed value is compared to the chi-square probability table to identify the p-value (degree of freedom is 1). The percentage of antibodies indicated in the double y-axis graph was computed by dividing the number of antibodies in the corresponding cell to the total number of antibodies (sum of cell a, b, c and d).

The number of articles corresponding to each human target protein was assessed by searching the NIH protein database (https://www.ncbi.nlm.nih.gov/protein/) on May 4, 2023.

## Resource information (alphabetical order)

| Name of the resource | RRID | Website |
| --- | --- | --- |
| Antibody Registry | RRID:SCR_006397 | https://antibodyregistry.org |
| Cancer Dependency Map Portal (DepMap) | RRID:SCR_017655 | https://depmap.org/portal/ |
| CiteAb | RRID:SCR_009653 | https://www.citeab.com |
| F1000research (YCharOS Gateway) | - | https://f1000research.com/ycharos |
| NIH protein database | - | https://www.ncbi.nlm.nih.gov/protein/ |
| Universal Protein Resource (Uniprot) | RRID:SCR_002380 | https://www.uniprot.org/ |
| ZENODO (YCharOS community) | - | https://zenodo.org/communities/ycharos |

## Acknowledgements

This work was supported by the Emory-Sage-SGC TREAT-AD center established by the National Institute on Aging (NIA) U54AG065187 grant and additional support by RF1AG057443, by a grant from the Michael J. Fox Foundation for Parkinson's Research (no. 18331), by a grant from the Motor Neurone Disease Association (UK), the ALS Association (USA) and ALS Society of Canada to develop ALS-RAP, by a Canadian Institutes of Health Research Foundation Grant (FDN154305) and by the Government of Canada through Genome Canada, Genome Quebec and Ontario Genomics (OGI-210). The Structural Genomics Consortium is a registered charity (no. 1097737) that receives funds from Bayer AG, Boehringer Ingelheim, Bristol Myers Squibb, Genentech, Genome Canada through Ontario Genomics Institute (grant no. OGI-196), the EU and EFPIA through the Innovative Medicines Initiative 2 Joint Undertaking (EUbOPEN grant no. 875510), Janssen, Merck KGaA (also known as EMD in Canada and the United States), Pfizer and Takeda. RA is supported by a Mitacs postdoctoral fellowship. AB is the co-founder and serves as the CEO of SciCrunch Inc, a company that works with publishers to improve the rigor and transparency of scientific manuscripts. Images were collected and/or image processing and analysis for this manuscript was performed in the McGill University Advanced BioImaging Facility (ABIF), RRID:SCR_017697. We thank Chetan Raina (YCharOS Inc) for his important contribution to the creation of an open scientific ecosystem of antibody manufacturers and knockout cell line suppliers.

# Additional information

## Funding

| Funder | Grant reference number | Author |
| --- | --- | --- |
| National Institute on Aging | U54AG065187 | Aled M Edwards |
| Michael J. Fox Foundation for Parkinson's Research | 18331 | Peter McPherson |
| ALS Society of Canada | | Thomas M Durcan<br>Aled M Edwards<br>Peter McPherson<br>Carl Laflamme |
| ALS Association | | Thomas M Durcan<br>Aled M Edwards<br>Peter McPherson<br>Carl Laflamme |
| Motor Neurone Disease Association | | Thomas M Durcan<br>Aled M Edwards<br>Peter McPherson<br>Carl Laflamme |
| Canadian Institutes of Health Research | FDN154305 | Peter McPherson |
| Genome Canada | OGI-210 | Peter McPherson<br>Carl Laflamme |
| Genome Quebec | OGI-210 | Peter McPherson<br>Carl Laflamme |
| Mitacs | Postdoctoral Fellowship | Riham Ayoubi |
| National Institute on Aging | RF1AG057443 | Aled M Edwards |
| Ontario Genomics | OGI-210 | Peter McPherson<br>Carl Laflamme |

The funders had no role in study design, data collection and interpretation, or the decision to submit the work for publication.

## Author contributions

Riham Ayoubi, Supervision, Investigation, Methodology; Joel Ryan, Software, Methodology; Michael S Biddle, Peter Eckmann, Anita Bandrowski, Data curation; Walaa Alshafie, Investigation, Methodology; Maryam Fotouhi, Vera Ruiz Moleon, Donovan Worrall, Ian McDowell, Investigation; Sara Gonzalez Bolivar, Resources, Investigation; Kathleen Southern, Thomas M Durcan, Claire Brown, Resources; Wolfgang Reintsch, Methodology; Harvinder Virk, Resources, Data curation, Writing – review and editing; Aled M Edwards, Conceptualization, Funding acquisition, Project administration, Writing – review and editing; Peter McPherson, Conceptualization, Resources, Supervision, Funding acquisition, Methodology, Project administration, Writing – review and editing; Carl Laflamme, Conceptualization, Resources, Data curation, Formal analysis, Supervision, Funding acquisition, Investigation, Methodology, Writing - original draft, Project administration, Writing – review and editing

## Author ORCIDs

Riham Ayoubi ⓘ https://orcid.org/0000-0001-6633-655X
Kathleen Southern ⓘ http://orcid.org/0000-0002-4125-3608
Harvinder Virk ⓘ http://orcid.org/0000-0002-9739-9593
Carl Laflamme ⓘ http://orcid.org/0000-0001-5906-025X

Reviewer #1 (Public Review): https://doi.org/10.7554/eLife.91645.2.sa1
Reviewer #2 (Public Review): https://doi.org/10.7554/eLife.91645.2.sa2
Author Response https://doi.org/10.7554/eLife.91645.2.sa3

# Additional files

## Supplementary files
• MDAR checklist

## Data availability

All data generated and analysed during this study are included in the corresponding antibody characterization reports openly available on the ZENODO open data repository (links to ZENODO reports are provided in *Figure 1—source data 1*).

The following datasets were generated:

| Author(s) | Year | Dataset title | Dataset URL | Database and Identifier |
|---|---|---|---|---|
| Fotouhi M, Ryan J, Reintsch W, Worrall D, Ayoubi R, Durcan TM, Brown CM, McPherson PS, Laflamme C | 2023 | Antibody Characterization Report for Alsin | https://doi.org/10.5281/zenodo.7671674 | Zenodo, 10.5281/zenodo.7671674 |
| Ayoubi R, Fotouhi M, Ryan J, Worrall D, Reintsch W, Durcan TM, Brown CM, McPherson PS, Laflamme C | 2023 | Antibody Characterization Report for Amyloid-beta precursor protein | https://doi.org/10.5281/zenodo.7971926 | Zenodo, 10.5281/zenodo.7971926 |
| Ayoubi R, Worrall D, McPherson PS, Laflamme C | 2023 | Antibody Characterization Report for Angiogenin | https://doi.org/10.5281/zenodo.7671286 | Zenodo, 10.5281/zenodo.7671286 |
| Alshafie W, Ayoubi R, Nicouleau M, Durcan TM, McPherson PS, Laflamme C | 2022 | Antibody Characterization Report for Annexin A11 | https://doi.org/10.5281/zenodo.5903684 | Zenodo, 10.5281/zenodo.5903684 |
| Ayoubi R, McPherson PS, Laflamme C | 2022 | Antibody Characterization Report for Apolipoprotein E | https://doi.org/10.5281/zenodo.7249055 | Zenodo, 10.5281/zenodo.7249055 |
| Alshafie W, Fotouhi M, Southern K, McPherson PS, Laflamme C | 2021 | Antibody Characterization Report for Ataxin-2 | https://doi.org/10.5281/zenodo.5061824 | Zenodo, 10.5281/zenodo.5061824 |
| Villegas L, Alshafie W, Fotouhi M, You Z, Durcan TM, McPherson PS, Laflamme C | 2021 | Antibody Characterization Report for Ataxin-3 | https://doi.org/10.5281/zenodo.5574175 | Zenodo, 10.5281/zenodo.5574175 |
| Ayoubi R, Alshafie W, Shapovalov I, Greer PA, McPherson PS, Laflamme C | 2021 | Antibody characterization report for Calpain-2 catalytic subunit | https://doi.org/10.5281/zenodo.5259215 | Zenodo, 10.5281/zenodo.5259215 |
| Ayoubi R, Alshafie W, McPherson PS, Laflamme C | 2021 | Antibody Characterization Report for CD44 antigen | https://doi.org/10.5281/zenodo.4730966 | Zenodo, 10.5281/zenodo.4730966 |
| Fotouhi M, Alshafie W, Shlaifer I, Ayoubi R, Durcan TM, McPherson PS, Laflamme C | 2022 | Antibody Characterization Report for Charged multivesicular body protein 2b | https://doi.org/10.5281/zenodo.6370501 | Zenodo, 10.5281/zenodo.6370501 |
| Ayoubi R, Alshafie W, Straub I, McPherson PS, Laflamme C | 2021 | Antibody Characterization Report for Coiled-coil-helix-coiled-coil-helix domain-containing protein 10, mitochondrial (CHCHD10) | https://doi.org/10.5281/zenodo.5259992 | Zenodo, 10.5281/zenodo.5259992 |

*Continued on next page*

*Continued*

| Author(s) | Year | Dataset title | Dataset URL | Database and Identifier |
|---|---|---|---|---|
| Ayoubi R, Alshafie W, You Z, Durcan TM, McPherson PS, Laflamme C | 2021 | Antibody Characterization Report for Dynamin-1 | https://doi.org/10.5281/zenodo.4724181 | Zenodo, 10.5281/zenodo.4724181 |
| Ayoubi R, Alshafie W, McPherson PS, Laflamme C | 2022 | Antibody Characterization Report for E3 ubiquitin-protein ligase Itchy homolog (Itch) | https://doi.org/10.5281/zenodo.6566970 | Zenodo, 10.5281/zenodo.6566970 |
| Ayoubi R, Alshafie W, Dorval G, Durcan TM, McPherson PS, Laflamme C | 2021 | Antibody Characterization Report for E3 ubiquitin-protein ligase parkin (Parkin) | https://doi.org/10.5281/zenodo.5747356 | Zenodo, 10.5281/zenodo.5747356 |
| Ayoubi R, Fotouhi M, Ryan J, Reintsch W, Durcan TM, Brown CM, McPherson PS, Laflamme C | 2022 | Antibody Characterization Report for Endothelin-converting enzyme 1 | https://doi.org/10.5281/zenodo.7459248 | Zenodo, 10.5281/zenodo.7459248 |
| Alshafie W, Ayoubi R, McPherson PS, Laflamme C | 2021 | Antibody Characterization Report for Equilibrative nucleoside transporter 1 SLC29A1 (ENT1) | https://doi.org/10.5281/zenodo.4733134 | Zenodo, 10.5281/zenodo.7324605 |
| Fotouhi M, You Z, Durcan TM, McPherson PS, Laflamme C | 2021 | Antibody Characterization Report for Gelsolin | https://doi.org/10.5281/zenodo.4724188 | Zenodo, 10.5281/zenodo.4724188 |
| Ayoubi R, Alshafie W, Dekakra-Bellili L, McPherson PS, Laflamme C | 2022 | Antibody Characterization Report for Hamartin | https://doi.org/10.5281/zenodo.6370607 | Zenodo, 10.5281/zenodo.6607513 |
| Ayoubi R, Fotouhi M, Ryan J, Reintsch W, Gonzalez Bolivar S, Durcan TM, Brown CM, McPherson PS, Laflamme C | 2023 | Antibody Characterization Report for Leucine-rich repeat kinase 2 (LRRK2) | https://doi.org/10.5281/zenodo.7971965 | Zenodo, 10.5281/zenodo.7987195 |
| Ayoubi R, McPherson PS, Laflamme C | 2022 | Antibody Characterization Report for Macrophage colony-stimulating factor 1 (CSF-1) | https://doi.org/10.5281/zenodo.6941512 | Zenodo, 10.5281/zenodo.6941512 |
| Ayoubi R, Alshafie W, McPherson PS, Laflamme C | 2021 | Antibody Characterization Report for Matrin-3 | https://doi.org/10.5281/zenodo.5644346 | Zenodo, 10.5281/zenodo.5644346 |
| Ayoubi R, McPherson PS, Laflamme C | 2021 | Antibody Characterization Report for Midkine | https://doi.org/10.5281/zenodo.5644321 | Zenodo, 10.5281/zenodo.5644321 |
| Ayoubi R, Alshafie W, Dekakra-Bellili L, McPherson PS, Laflamme C | 2022 | Antibody Characterization Report for Mitogenactivated protein kinase 1 (MAPK1) | https://doi.org/10.5281/zenodo.6941499 | Zenodo, 10.5281/zenodo.6941499 |
| Alshafie W, Ayoubi R, Fotouhi M, McPherson PS, Laflamme C | 2021 | Antibody Characterization Report for Moesin | https://doi.org/10.5281/zenodo.4724169 | Zenodo, 10.5281/zenodo.4627263 |
| Ayoubi R, Alshafie W, McPherson PS, Laflamme C | 2022 | Antibody Characterization Report for NADH dehydrogenase [ubiquinone] iron-sulfur protein 2 (NDUFS2) | https://doi.org/10.5281/zenodo.5903708 | Zenodo, 10.5281/zenodo.5903708 |

*Continued on next page*

*Continued*

| Author(s) | Year | Dataset title | Dataset URL | Database and Identifier |
|---|---|---|---|---|
| Ayoubi R, You Z, Durcan TM, McPherson PS, Laflamme C | 2022 | Antibody Characterization Report for Neurosecretory protein VGF | https://doi.org/10.5281/zenodo.5903141 | Zenodo, 10.5281/zenodo.5903141 |
| Ayoubi R, Fotouhi M, Ryan J, Reintsch W, Durcan TM, Brown CM, McPherson PS, Laflamme C | 2022 | Antibody Characterization Report for QPRTase (Nicotinate-nucleotide pyrophosphorylase [carboxylating]) | https://doi.org/10.5281/zenodo.7459387 | Zenodo, 10.5281/zenodo.7459387 |
| Fotouhi M, Ryan J, Worrall D, Ayoubi R, Reintsch W, Durcan TM, Brown CM, McPherson PS, Laflamme C | 2023 | Antibody Characterization Report for RNA-binding protein TIA1 | https://doi.org/10.5281/zenodo.7671718 | Zenodo, 10.5281/zenodo.7671718 |
| Alshafie W, Fotouhi M, Shlaifer I, Durcan TM, McPherson PS, Laflamme C | 2021 | Antibody Characterization Report for Optineurin | https://doi.org/10.5281/zenodo.4730992 | Zenodo, 10.5281/zenodo.4730992 |
| Ayoubi R, Fotouhi M, You Z, Durcan TM, McPherson PS, Laflamme C | 2021 | Antibody Characterization Report for Peroxiredoxin-1 | https://doi.org/10.5281/zenodo.4818397 | Zenodo, 10.5281/zenodo.4818397 |
| Ayoubi R, Alshafie W, Nicouleau M, Durcan TM, McPherson PS, Laflamme C | 2021 | Antibody Characterization Report for Peroxiredoxin-6 | https://doi.org/10.5281/zenodo.4730953 | Zenodo, 10.5281/zenodo.4730953 |
| Ayoubi R, Fotouhi M, Moleon RuizV, Ryan J, Worrall D, Reintsch W, Durcan TM, Brown CM, McPherson PS, Laflamme C | 2023 | Antibody Characterization Report for Plasma membrane calcium-transporting ATPase 1 | https://doi.org/10.5281/zenodo.7971932 | Zenodo, 10.5281/zenodo.7971932 |
| Ayoubi R, Fotouhi M, You Z, Durcan TM, McPherson PS, Laflamme C | 2021 | Antibody Characterization Report for Plectin | https://doi.org/10.5281/zenodo.4724176 | Zenodo, 10.5281/zenodo.4724176 |
| Ayoubi R, Gonzalez Bolivar S, McPherson PS, Laflamme C | 2022 | Antibody Characterization Report for Pleiotrophin | https://doi.org/10.5281/zenodo.7459312 | Zenodo, 10.5281/zenodo.7987214 |
| Ayoubi R, Alshafie W, McPherson PS, Laflamme C | 2022 | Antibody Characterization Report for Pro-cathepsin H | https://doi.org/10.5281/zenodo.5903713 | Zenodo, 10.5281/zenodo.5903713 |
| McDowell I, Ayoubi R, Ryan J, Fotouhi M, Reintsch W, Durcan TM, Brown CM, McPherson PS, Laflamme C | 2022 | Antibody Characterization Report for Profilin-1 | https://doi.org/10.5281/zenodo.7249258 | Zenodo, 10.5281/zenodo.7249258 |
| Ayoubi R, Fotouhi M, Ryan J, Gonzalez Bolivar S, Worrall D, Reintsch W, Durcan TM, Brown CM, McPherson PS, Laflamme C | 2023 | Antibody Characterization Report for Prolow-density lipoprotein receptor-related protein1 (LRP-1) | https://doi.org/10.5281/zenodo.7971951 | Zenodo, 10.5281/zenodo.7971951 |

*Continued on next page*

*Continued*

| Author(s) | Year | Dataset title | Dataset URL | Database and Identifier |
|---|---|---|---|---|
| Ayoubi R, Alshafie W, Fotouhi M, McPherson PS, Laflamme C | 2022 | Antibody Characterization Report for Retinoic acid receptor RXR-alpha | https://doi.org/10.5281/zenodo.6566983 | Zenodo, 10.5281/zenodo.6566983 |
| Worrall D, Ryan J, Ayoubi R, Fotouhi M, Reintsch W, Durcan TM, Brown C, McPherson PS, Laflamme C | 2022 | Antibody Characterization Report for Rho GDP-dissociation inhibitor 1 (Rho GDI 1) | https://doi.org/10.5281/zenodo.7249083 | Zenodo, 10.5281/zenodo.7249221 |
| Alshafie W, Fotouhi M, You Z, Durcan TM, McPherson PS, Laflamme C | 2021 | Antibody Characterization Report for RNA-binding protein FUS | https://doi.org/10.5281/zenodo.5259944 | Zenodo, 10.5281/zenodo.5259945 |
| Ayoubi R, McPherson PS, Laflamme C | 2022 | Antibody Characterization Report for Secreted frizzled-related protein 1 | https://doi.org/10.5281/zenodo.6370454 | Zenodo, 10.5281/zenodo.6370454 |
| Ayoubi R, Alshafie W, Shlaifer I, Durcan TM, McPherson PS, Laflamme C | 2021 | Antibody Characterization Report for Sequestosome-1 | https://doi.org/10.5281/zenodo.4818440 | Zenodo, 10.5281/zenodo.4818440 |
| Ayoubi R, McPherson PS, Laflamme C | 2022 | Antibody Characterization Report for Serine protease HTRA1 | https://doi.org/10.5281/zenodo.7249120 | Zenodo, 10.5281/zenodo.7986850 |
| Alshafie W, Fotouhi M, Shlaifer I, You Z, Durcan TM, McPherson PS, Laflamme C | 2021 | Antibody Characterization Report for Serine/threonine-protein kinase Nek1 | https://doi.org/10.5281/zenodo.5061736 | Zenodo, 10.5281/zenodo.5061736 |
| Alshafie W, Fotouhi M, Shlaifer I, Durcan TM, McPherson PS, Laflamme C | 2021 | Antibody Characterization Report for Serine/threonine-protein kinase TBK1 | https://doi.org/10.5281/zenodo.5061682 | Zenodo, 10.5281/zenodo.6402968 |
| Alshafie W, Fotouhi M, Ayoubi R, Nicouleau M, Durcan TM, McPherson PS, Laflamme C | 2021 | Antibody Characterization Report for Sigma non-opioid intracellular receptor 1 | https://doi.org/10.5281/zenodo.5644356 | Zenodo, 10.5281/zenodo.5644356 |
| Ayoubi R, Ryan J, Dekakra-Bellili L, Brown CM, McPherson PS, Laflamme C | 2022 | Antibody Characterization Report for Signal transducer and activator of transcription 5B (STAT5B) | https://doi.org/10.5281/zenodo.7249185 | Zenodo, 10.5281/zenodo.7249185 |
| Ayoubi R, Nicouleau M, Durcan TM, McPherson PS, Laflamme C | 2022 | Antibody Characterization Report for SPARC-related modular calcium-binding protein 1 (SMOC1) | https://doi.org/10.5281/zenodo.6566878 | Zenodo, 10.5281/zenodo.6566878 |
| Alshafie W, Ayoubi R, McPherson PS, Laflamme C | 2021 | Antibody Characterization Report for Spastin | https://doi.org/10.5281/zenodo.5644358 | Zenodo, 10.5281/zenodo.5644358 |
| Ayoubi R, Fotouhi M, Ryan J, Reintsch W, Durcan TM, Brown CM, McPherson PS, Laflamme C | 2022 | Antibody Characterization Report for Spatacsin | https://doi.org/10.5281/zenodo.7459431 | Zenodo, 10.5281/zenodo.7459431 |

*Continued on next page*

*Continued*

| Author(s) | Year | Dataset title | Dataset URL | Database and Identifier |
|-----------|------|---------------|-------------|-------------------------|
| Ayoubi R, Alshafie W, You Z, Durcan TM, McPherson PS, Laflamme C | 2021 | Antibody Characterization Report for Superoxide dismutase [Cu-Zn] (SOD1) | https://doi.org/10.5281/zenodo.5061103 | Zenodo, 10.5281/zenodo.5061103 |
| Ayoubi R, Alshafie W, Fotouhi M, You Z, Durcan TM, McPherson PS, Laflamme C | 2021 | Antibody Characterization Report for Synaptotagmin-1 | https://doi.org/10.5281/zenodo.5644331 | Zenodo, 10.5281/zenodo.5644331 |
| Ayoubi R, McPherson PS, Laflamme C | 2022 | Antibody Characterization Report for Syndecan-4 | https://doi.org/10.5281/zenodo.6566857 | Zenodo, 10.5281/zenodo.6566857 |
| Ayoubi R, Alshafie W, McPherson PS, Laflamme C | 2022 | Antibody Characterization Report for Syntaxin-4 | https://doi.org/10.5281/zenodo.5903087 | Zenodo, 10.5281/zenodo.5903087 |
| Worrall D, Ryan J, Fotouhi M, Ayoubi R, Reintsch W, Durcan TM, Brown CM, McPherson PS, Laflamme C | 2022 | Antibody Characterization Report for TDP-43 | https://doi.org/10.5281/zenodo.6841232 | Zenodo, 10.5281/zenodo.7249802 |
| Fotouhi M, Moleon RuizV, Ryan J, Worrall D, Ayoubi R, Reintsch W, Durcan TM, Brown CM, McPherson PS, Laflamme C | 2023 | Antibody Characterization Report for Valosin-containing protein VCP | https://doi.org/10.5281/zenodo.7971904 | Zenodo, 10.5281/zenodo.7971904 |
| Ayoubi R, Fotouhi M, Ryan J, Reintsch W, Durcan TM, Brown CM, McPherson PS, Laflamme C | 2022 | Antibody Characterization Report for Transmembrane protein 106B | https://doi.org/10.5281/zenodo.7459629 | Zenodo, 10.5281/zenodo.7459629 |
| Ayoubi R, Alshafie W, Dekakra-Bellili L, McPherson PS, Laflamme C | 2022 | Antibody Characterization Report for Tuberin | https://doi.org/10.5281/zenodo.6370612 | Zenodo, 10.5281/zenodo.6377409 |
| Fotouhi M, Alshafie W, Shlaifer I, Ayoubi R, Durcan TM, McPherson PS, Laflamme C | 2022 | Antibody Characterization Report for Tubulin alpha-4A chain | https://doi.org/10.5281/zenodo.5903719 | Zenodo, 10.5281/zenodo.7987237 |
| Alshafie W, Fotouhi M, Ayoubi R, McPherson PS, Laflamme C | 2022 | Antibody Characterization Report for TYRO protein tyrosine kinase-binding protein (TYROBP) | https://doi.org/10.5281/zenodo.6941517 | Zenodo, 10.5281/zenodo.6941517 |
| Alshafie W, Fotouhi M, Ayoubi R, McPherson PS, Laflamme C | 2022 | Antibody Characterization Report for Tyrosine-protein kinase SYK | https://doi.org/10.5281/zenodo.6566940 | Zenodo, 10.5281/zenodo.6566940 |
| McDowell I, Fotouhi M, Ryan J, Ayoubi R, Reintsch W, Durcan TM, Brown CM, McPherson PS, Laflamme C | 2022 | Antibody Characterization Report for Ubiquilin-2 | https://doi.org/10.5281/zenodo.7459541 | Zenodo, 10.5281/zenodo.7459541 |

*Continued*

| Author(s) | Year | Dataset title | Dataset URL | Database and Identifier |
|---|---|---|---|---|
| Ayoubi R, Fotouhi RM, Ryan J, Reintsch W, Durcan TM, Brown CM, McPherson PS, Laflamme C | 2023 | Antibody Characterization Report for hVPS35 (Vacuolar protein sorting-associated protein 35) | https://doi.org/10.5281/zenodo.7671730 | Zenodo, 10.5281/zenodo.7671730 |
| Fotouhi M, Alshafie W, Shlaifer I, Ayoubi R, Durcan TM, McPherson PS, Laflamme C | 2022 | Antibody Characterization Report for Vesicle-associated membrane protein-associated protein B/C (VAPB) | https://doi.org/10.5281/zenodo.6370535 | Zenodo, 10.5281/zenodo.6370535 |

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
