## [Editor Report · eLife assessment]

Antibodies are some of the most important tools in biomedical research. However, their quality and specificity vary significantly. This **fundamental** study provides guidelines for how the quality of an antibody should be assessed and recorded and provides **compelling** data on the selected antibodies. This paper will be of interest to researchers working in experimental cell biology.

---

## [Referee Report · Reviewer #1 (Public Review)]

The research addresses a key problem in life sciences: While there are millions of commercially available antibodies to human proteins, researchers often find that the reagents do not perform in the assays they are specified for. The consequence is wasted time and research funding, and the publication of misleading results.

Manufacturers' catalogues often contain images of western blots. Researchers are likely to select antibodies that stain a single band at the position expected from the mass of the intended target. However, the good results shown in catalogues are often not reproduced when researchers use the antibody in their own laboratories. A single band is also weak evidence since many proteins have similar mass and because assessment of mass by WB is at best approximate. In addition, results obtained by WB may not predict performance in applications where the antibody is used to recognize folded proteins. Examples include immunoprecipitation (IP) of native proteins in cell lysates and staining of viable or formalin-fixed/permeablized cells for flow cytometry or immunofluorescence microscopy (IF).

The authors of this manuscript are from the Canadian, public interest open-science company YCharos. The company webpage (ycharos.com) explains that they have partnered with many leading manufacturers of research antibodies and that their mission is to characterize commercially available antibody reagents for every human protein.

The authors have developed a standardized pipeline where antibodies are used in WB, IP of native proteins from cell lysates (WB readout) and IF (staining of cell lines that have been fixed with paraformaldehyde and permeabilized with Triton x100). A key component is the use of knockout cell lines as negative controls in WB and IF. Eight cell lines were selected as positive controls on the basis of mRNA expression data that are publicly available in the Expression 22Q1 database.

Reports for antibodies to each protein are made available online at https://ZENODO.org/communities/ycharos/ as images of western blots, and immunofluorescence staining. In addition, reports for each target are available at https://ycharos.com/data/ .

MANUSCRIPT:

The manuscript describes validation criteria and results obtained with 614 commercially available antibodies to 65 proteins relevant for neuroscience A major achievement is the identification of successful renewable antibodies for 50/56 (77%) proteins in WB, 49/65 (75%) for IP and (54%) for IF. There can be little doubt that the approach represents a gold standard in antibody validation. The manuscript therefore represents a guide to a very valuable resource that should be of considerable interest to the scientific community.

While the results are convincing, they could be more accessible. In the current format, researchers have to download reports for each target and look through all images to identify the most useful antibodies from the images. The reports I reviewed did not draw conclusions on performance. A searchable database that returns validated antibodies for each application seems necessary.

It is worth noting that 95% of the tested antibodies were specified by the manufacturer for use in WB. This supports the view that manufacturers use WB as a first-pass test (Nat Methods. 2017 Feb 28;14(3):215) and that most commercial antibodies are developed to recognize epitopes that are exposed in unfolded proteins. Important exceptions are those used for ELISA or staining of viable cells for flow cytometry. 44% of antibodies specified for WB were classified as "successful" meaning a single band that was absent in the negative control (knockout/KO lysate). Another 35% detected the intended target but showed additional bands that were present also in the KO lysate. A key question is to what extent off-target binding was predictable from the WBs provided by the manufacturers. Thus, how often did the authors find multiple bands when the catalogue image showed a single band and vice versa?

The authors correctly point out that manufacturers rarely test their reagents in IP. Thus, there is little information about antibodies capable of binding folded proteins. It is encouraging that as many as 37% of those not specified for IP were able to enrich their targets from cell lysates. Yet it is important to explain that a test that involves readout by WB provides information about on-target binding only. Cross-reactive proteins will generally not be detected when blots are stained with an antibody reactive with a different epitope than the one used for IP. Possible solutions to overcome this limitation such as the use of mass spectrometry as readout should be discussed (Nature Methods volume 12, pages 725-731 2015).

Performance in immunofluorescence microscopy was performed on cells that were fixed in 4% paraformaldehyde and then permeabilized with 0.1% Triton-X100. It seems reasonable to assume that this treatment mainly yields folded proteins wherein some epitopes are masked due to cross-linking. The expectation is therefore that results from IP are more predictive for on-target binding in IF than are WB results (Nature Methods volume 12, pages725-731 2015). It is therefore surprising that IP and WB were found to have similar predictive value for performance in IF (supplemental Fig. 3). It would be useful to know if failure in IF was defined as lack of signal, lack of specificity (i.e. off-target binding) or both. Again, it is important to note the IP/western protocol used here does not test for specificity.

The authors report that recombinant antibodies perform better than standard monoclonals/mAbs or polyclonal antibodies. Again, a key question is to what extent this was predictable from the validation data provided by the manufacturers. It seems possible that the recombinant antibodies submitted by the manufacturers had undergone more extensive validation than standard mAbs and polyclonals.

Overall, the manuscript describes a landmark effort for systematic validation of research antibodies. The results are of great importance for the very large number of researchers who use antibodies in their research. The main limitations are the high cost and low throughput. While thorough testing of 614 antibodies is impressive and important, the feasibility of testing hundreds of thousands of antibodies on the market should be discussed in more detail.

---

## [Referee Report · Reviewer #2 (Public Review)]

The paper nicely demonstrates the extent of the issue with the unreliability of commercial antibodies and describes a highly significant initiative for the robust validation of antibodies and recording this data so that others can benefit. It is a great idea to have all individual antibody characterisation reports available on Zenodo - these reports are comprehensive, clear and available to everyone.

A significant proportion of all life science research conclusions are based on data obtained through the use of antibodies. The quality and specificity of antibodies vary significantly. Until now there has been no uniform generally recognised approach to how to systematically assess and rate antibody specificity and quality. Furthermore, the applications that a particular antibody can be used in including western blot, immunofluorescence or immunoprecipitation are frequently not known. This paper provides important guidelines for how the quality of an antibody should be assessed and recorded and data made freely available via a Zenodo repository. This study will ensure that researchers only use well-validated antibodies for their work. A worrying aspect of this paper is that many poor-quality antibodies that failed validation are reportedly being widely used in the literature. More than 60% of all antibodies recommended for immunofluorescence failed QC. This study will have broad interest. I would recommend that all researchers select their antibodies using the database described in the paper and follow its recommendations for how antibodies should be thoroughly validated before being used in research. Hopefully, other researchers can contribute to this database in the future all widely used antibodies will eventually be well characterized. This should improve the quality and reproducibility of life science research.

---

## [Author Response]

**Reviewer #1:**

We thank Reviewer #1 for their review of our manuscript.

Reviewer #1, comment #1: “The authors of this manuscript are from the Canadian, public interest open-science company YCharos.”.

It is important to state that none of the authors work for YCharOS. The YCharOS company has created an open ecosystem consisting of antibody manufacturers, knockout cell lines providers, academics, granting agencies and publishers. The Antibody Characterization Group (participating authors are affiliated to the Department of Neurology and Neurosurgery, Structural Genomics Consortium, The Montreal Neurological Institute, McGill University) works in collaboration with YCharOS to have access to commercial antibodies and knockout cell lines donated by YCharOS’ manufacturer partners.

Reviewer #1, comment #2: In regard to ZENODO antibody characterization reports prepared by this group, Reviewer #1 wrote: “While the results are convincing, they could be more accessible. In the current format, researchers have to download reports for each target and look through all images to identify the most useful antibodies from the images. The reports I reviewed did not draw conclusions on performance. A searchable database that returns validated antibodies for each application seems necessary.”

After careful consideration and consultation with YCharOS industry partners, we decided not to rate the performance of the antibodies tested. It was determined that antibody selection is best left to the user, who should analyze all parameters, including the type of antibody to be chosen (recombinant-monoclonal, recombinant-polyclonal, monoclonal), the species used to generate the antibody, the species predicted to react with the antibody, performance in a specific application, antigen sequences, and antibody cost.

Reviewer #1, comment #3: “A key question is to what extent off-target binding was predictable from the WBs provided by the manufacturers. Thus, how often did the authors find multiple bands when the catalogue image showed a single band and vice versa?”

In many cases, the antibodies were tested on cell lines other than those used by the manufacturers. Given that protein expression is specific to each line, we can't answer this question properly.

Reviewer #1, comment #4: “Cross-reactive proteins will generally not be detected when blots are stained with an antibody reactive with a different epitope than the one used for IP. Possible solutions to overcome this limitation such as the use of mass spectrometry as readout should be discussed (Nature Methods volume 12, pages 725- 731 2015)”.

Our protocols only inform whether an antibody can capture the intended target, without any evaluation of the extend to the capture of unwanted, cross-reactive proteins. Thus, our data can only be used to aid in selection of the best performing antibodies for IP – our data does not inform profiling of non-specific interactions.

IP/mass spec is an excellent approach for evaluating antibody performance for IP, and authors on this manuscript are experts in proteomics and recognize the importance of this methodology. We have considered implementing IP/mass in our platform. However, there are limitations, such as the cost of the approach and the difficulty of detecting smaller proteins or proteins with a certain amino acid composition (high presence of Cys, Arg or Lys). Fundamentally, we have decided to focus on throughput relative to details in this regard.

Reviewer #1, comment #5: “Performance in immunofluorescence microscopy was performed on cells that were fixed in 4% paraformaldehyde and then permeabilized with 0.1% Triton-X100. It seems reasonable to assume that this treatment mainly yields folded proteins wherein some epitopes are masked due to cross-linking. The expectation is therefore that results from IP are more predictive for on-target binding in IF than are WB results (Nature Methods volume 12, pages725-731 2015). It is therefore surprising that IP and WB were found to have similar predictive value for performance in IF (supplemental Fig. 3). It would be useful to know if failure in IF was defined as lack of signal, lack of specificity (i.e. off-target binding) or both. Again, it is important to note the IP/western protocol used here does not test for specificity.”

The assessment of antibody performance is biased by how antibodies were originally tested by suppliers. Manufacturers primarily validate their antibody by WB. Thus, most antibodies immunodetect their intended target for WB. Thus, in retrospect, we tested a biased pool of antibodies that detect linear epitopes. Still, we observed that a large cohort of antibodies show specificity for their target across all three applications or for specific combinations of applications. This slightly challenges the idea that antibodies are fit-for-purpose reagents and can recognize either linear or native epitopes - a significant number of antibodies can specifically detect both types of epitope.

Reviewer #1, comment #6: “The authors report that recombinant antibodies perform better than standard monoclonals/mAbs or polyclonal antibodies. Again, a key question is to what extent this was predictable from the validation data provided by the manufacturers. It seems possible that the recombinant antibodies submitted by the manufacturers had undergone more extensive validation than standard mAbs and polyclonals”.

Our antibody manufacturing partners indicated that the recombinant antibodies are more recent products and have been more extensively characterized relative to standard polyclonal or monoclonal antibodies.

The main message is that recombinant antibodies can be used in all applications once validated. Although recombinant antibodies are available for many proteins, the scientific community is not adopting these renewable regents as we believe it should. We hope that the data provided will encourage scientists to adopt recombinant technologies when available to improve research reproducibility.

Reviewer #1, comment #7: “Overall, the manuscript describes a landmark effort for systematic validation of research antibodies. The results are of great importance for the very large number of researchers who use antibodies in their research. The main limitations are the high cost and low throughput. While thorough testing of 614 antibodies is impressive and important, the feasibility of testing hundreds of thousands of antibodies on the market should be discussed in more detail.”

We thank the reviewer for this comment. One of our challenges is to increase the platform's throughput to succeed in our mission to characterize antibodies for all human gene products. We will continue to test antibodies using protocols agreed upon with our partners, commonly used in the laboratory, to ensure that ZENODO reports can serve as a guide to the wider community.

In terms of development our marketing efforts have been substantially accelerated by our new partnership with the journal F1000. We have begun to convert our reports into peer-reviewed papers (20 ZENODO reports were converted into F1000 articles). This conversion allows researchers to find our work via PubMed, and easily cite any study. Producing peer-reviewed articles also further enhances the credibility of our research and our project as a whole: https://f1000research.com/ycharos

Colleagues have published a letter to Nature explaining the problem and our technology platform: (Kahn, et al., Nature, 2023, DOI: https://doi.org/10.1038/d41586-023-02566-w).

This project has been presented worldwide, with a presence at major antibody conferences, such as the annual Antibody Validation meeting in Bath (PSM attended the meeting in September 2023). The authors are organizing a sponsored mini-symposium on antibody validation at the next American Society for Cell Biology (ASCB) meeting in December 2023 (Boston, USA): https://plan.core- apps.com/ascbembo2023/event/6fb928f06b0d672e088c6fa88e4d77fb

Colleagues have prepared petitions addressed to various governmental organizations (US, Canada, UK) to support characterization and validation of renewable antibodies: https://www.thesgc.org/news/support- characterization-and-validation-renewable-antibodies.

**Reviewer #2**

We thank Reviewer #2 for the review of the antibody characterization reports we have uploaded to ZENODO. A manuscript describing the full standard operating procedures of the platform, which has been used in all reports is in preparation, and should be available on a preprint server before the end of the year. Our protocols were reviewed and approved by each of YCharOS' manufacturer partners. Moreover, a recent editorial describes the platform used here and gives advice on how to interpret the data: https://doi.org/10.12688/f1000research.141719.1

Reviewer #2, comment #1: “A discussion of how the working concentrations of antibodies are selected and validated is required. Based on the dilutions described in the reports, it seems that dilutions suggested by the manufacturer were used - For LRRK2 it seems that antibody concentrations ranging from 0.06 to over 5 µg/ml for WB were used. Often commercial antibody comes in a BSA-containing buffer making it hard to validate the concentration of the antibody claimed by the manufacturer”.

The concentration recommended by the manufacturer is our starting point. For WB, when the signal is at the level of detectability, we will repeat with a ~5-10 fold increase in antibody concentration. For >80% of the antibody tested, the use of the recommended concentration led to the detection of bands (specific or not to the target protein).

Reviewer #2, comment #2: “In the authors' experience are the manufacturer's concentrations reliable? Additionally, if the information regarding applications provided by the manufacturers is unreliable how do the authors suggest working concentrations for antibodies to be assessed”?

We do not evaluate the concentration of antibodies internally. In the immunoprecipitation experiments, we use 2.0 µg of antibody for each IP, based on the concentration provided by the manufacturers. On Ponceau staining of membranes, we can observe the heavy and light chains of the primary antibodies used, giving an indication of the amount of antibodies added to the cell lysate. In most cases, the intensity of the heavy and light chains is comparable.

Reviewer #2, comment #3: “We understand that it would not be feasible to test every antibody at different concentrations, but this is an issue that should at least be mentioned. An antibody might be put in the wrong performance category solely because of the wrong concentration being used. Ie if an excellent antibody is used at too high a concentration, it may detect non-specific proteins that are not seen at lower dilutions where the antibody still picks up the desired antigen well”.

We agree with Reviewer #2, we do not use an optimal concentration for all tested antibodies. As mentioned previously, the concentration recommended by the manufacturer is our starting point. By testing multiple antibodies side-by-side against a single target protein, we can generally identify one or more specific and selective antibodies. We leave it to users of our reports to optimize the antibody concentration to suit their experimental needs.

Reviewer #2, comment #4: “Do the authors check different WB conditions ie 2h primary antibody with BSA or milk vs. overnight at 4 degrees with BSA or Milk”?

All primary antibodies are always tested in milk overnight at 4 degrees. The overnight incubation is convenient in the timeline of the protocol. All protocols were agreed upon after careful consultation with our partners.

Reviewer #2, comment #5: “Do the authors provide detailed WB protocols that include the description of the electrophoresis and type of gels used, transfer buffer and transfer method and time used, and conditions for all the primary and secondary blotting including times, buffers and dilutions of all antibodies and other reagents”?

This information is included in all ZENODO reports.

Reviewer #2, comment #6: “Do the authors discuss detection approaches- we have noticed for some antibodies there are significant different results using LICOR, ECL and other detection methods, with certain especially weaker antibodies preferring ECL-based methods”.

We only use ECL-based methods.

Reviewer #2, comment #7: “For IPs the amount of antibody needed can also vary-for some we can use 1 microgram or less, but for others, we need 5 to 10 micrograms. The amount of antibody needed to get maximal IP should be stated”.

We use 2.0 ug of antibodies and we have found this to be adequate for lower abundance proteins (e.g. Parkin - https://zenodo.org/records/5747356) and higher abundance proteins (e.g. PRDX6 - https://zenodo.org/records/4730953). Abundance is based on PaxDb.com. For Parkin and PRDX6, we were able to enrich the expected target in the IP and observe depletion in the unbound fraction. Optimization of the IP conditions is left to the antibody users.

Reviewer #2, comment #8: “Doing IPs with commercial antibodies can be very expensive or infeasible if many micrograms are needed especially if only packages of 10 micrograms for several hundred dollars are provided”.

This is a major advantage of the side-by-side comparison: the reader is free to choose between high-performance antibodies from different manufacturers, with varying antibody costs. We also work in partnership with the Developmental Studies Hybridoma Band (DSHB), which supplies antibodies on a cost recovery basis.

Reviewer #2, comment #9: “For IPs it is important to determine the percentage of antigen that is depleted from the supernatant for each IP. We think that this should be calculated and recorded in the Zenodo data. Some antibodies will only IP 10% of antigen whereas others may do 50% and others 80-90%. One rarely sees 100% depletion. For IPs the buffer detergent and salt concentration might also strongly influence the degree of IP and therefore these should be clearly stated”.

In Box 1, we define criteria of success. For IP, “under the conditions used, a successful primary antibody immunocaptures the target protein to at least 10% of the starting material”. Colleagues have written an editorial on how to interpret and analyze antibody performance https://f1000research.com/articles/12-1344.

The cell lysis buffer is a critical reagent when considering IP experiments. We use a commercial buffer consisting of 25 mM Tris-HCl pH 7.4, 150 mM NaCl, 1 mM EDTA, 1% NP-40 and 5% glycerol (Thermo Fisher, cat. #87787). This buffer is efficient to extract the target proteins we have studied thus far.

Reviewer #2, comment #10: “Whether antibodies cross-react with human, mouse and other species of antigens is always a major question. It is always good to test human and mouse cell lines if possible. If antibodies cross-react in WB, in the authors' experience will they also cross-react for IF and IP”?

The authors started this initiative by focusing on the 20,000 human proteins, defining an end point. We and our collaborators found that most of the cherry-picked selective antibodies for WB for human proteins, which manufacturers claim react with the murine version of the target proteins, were selective for murine tissue lysates.

Indeed, poorly performing antibodies in WB mostly failed IF and IP. However, selective antibodies for IF or specific for IP were generally (>90%) selective for WB.

Reviewer #2, comment #11: “Cell lines express proteins at vastly different levels and it is possible that the selected cell line does not express the antigen or expresses it at very low levels - this could be a reason for wrongly assessing an antibody not working. It would be useful to use cell lines in which MS data has defined the copy number of protein per cell and this figure could be included in the antibody data if available. This MS data is available for the vast majority of commonly used cells”.

We agree with Reviewer #2 that MS data are useful for target protein selection. At the moment, our approach using transcriptomic data provided on DepMap.org proved to be a successful mechanism for cell line selection. We have identified a specific antibody for WB for each target, enabling the validation of expression in the cell line selected.

For some protein targets, the parental line corresponding to the only commercial or academic knockout line available has weak protein expression. We thus needed to generate a KO clone in a second cell line background with high expression, and indeed found that some antibodies which failed in the first commercial line were successful in the new higher-expressing line (e.g CHCHD10 - https://zenodo.org/records/5259992).

Reviewer #2, comment #12: “Some proteins are glycosylated, ubiquitylated or degraded rapidly making them hard to see in WB analysis”.

We used the full gel/membrane length when analyzing antibody performance by WB. Indeed, proteins can show different isoforms and molecular weights compared to that based on amino acid sequence (e.g. SLC19A1 -https://zenodo.org/records/7324605).

Reviewer #2, comment # 13: “We have occasionally had proteins that appear unstable when heated with SDS- sample buffer before WB. For these, we still use SDS-Sample buffer but omit the heating step. I often wonder how necessary the heating step is”.

For WB, samples are heated to 65 degrees, then spun to remove any precipitate.

Reviewer #2, comment # 14: “For IF the methods by which cells are fixed and stained, and the microscope and settings, can significantly influence the final result. It would be important to carefully record all the methods and the microscope used”.

We agree with Reviewer #2 that many parameters influence antibody performance for imaging purposes. We are progressively implementing the OMERO software to monitor any experimental parameters and information (metadata) about the microscope itself.

Reviewer #2, comment # 15: “How do the authors recommend antibodies are stored? These should be very stable, but I have had reports from the lab that some antibodies become less good when stored and others that recommend storing at 4 degrees”.

Antibodies are aliquoted to avoid freeze-thaw cycles and stored at -20 degrees. If it is recommended to store antibodies at 4 degrees, we add glycerol to a final concentration of 50% and store them at -20 degrees.

Reviewer #2, comment # 16: “Would other researchers not part of the authors' team, be able to add their own data to this database validating or de-validating antibodies? This would rapidly increase the number of antibodies for which useful data would be available for. It would be nice to greatly expand the number of antibodies being used in research and this is not feasible for a single team to undertake”.

Yes! We believe that only a community effort can resolve the antibody liability crisis. We partner with the Antibody Registry (antibodyregistry.org - led by co-author Anita Bandrowski). In the Registry, each antibody is labelled with a unique identifier, and third-party validation information can be easily tagged to any antibody. Antibody users are invited to upload information about an antibody they have characterized into the Registry.